# E²I-VRWKV: Explicit EPI-Representation and Interaction-Aware Vision-RWKV for Light Field Semantic Segmentation

**Wei Zhang** [1 2]   **Chen Jia** [1 2]   **Xu Cheng** [1 2]   **Fan Shi** [1 2]   **Hui Liu** [1 2]   **Shengyong Chen** [1 2]

## Abstract

Pixel-level semantic segmentation of 4D light field (LF) data remains a considerable challenge, primarily due to the conflict between modeling complex spatial-angular dependencies and maintaining linear computational efficiency. Current linear models like VRWKV offer scalability but often fail to capture intrinsic geometric structures, leading to the structural collapse of Epipolar Plane Image (EPI) cues. To overcome these limitations, we propose E²I-VRWKV, an explicit EPI-Representation and Interaction-aware network that generates high-quality segmentation maps by embedding explicit geometric priors into a linear-complexity backbone. Specifically, we introduce the Light Field Epipolar-Aware Cross-Modal Attention (LF-ECMA) block. The key innovation lies in the integration of an EPI Geometric Prior Generator, which explicitly extracts disparity-sensitive biases to enforce geometric consistency, and a Geometric-Context Gating (GC-Gate) mechanism. This mechanism functions as a geometrically modulated aperture to dynamically calibrate the fusion of spatial and angular manifolds. Experiments on the UrbanLF benchmark demonstrate that our method outperforms other state-of-the-art (SOTA) methods, achieving 86.55% mIoU on UrbanLF-Real while maintaining an improved balance between accuracy and linear efficiency.

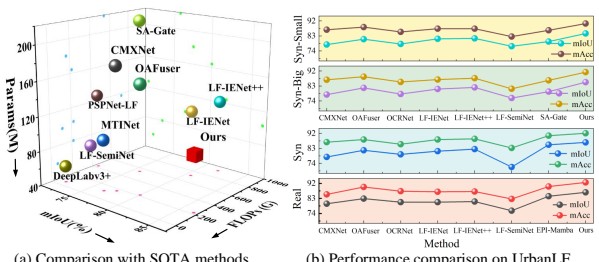

(a) Comparison with SOTA methods.    (b) Performance comparison on UrbanLF.

*Figure 1.* Efficiency and Performance of E²I-VRWKV on UrbanLF dataset. (a) Comparison with SOTA methods. (b) Comparison of four different benchmarks on performance, with normalized metrics; mIoU(%), mACC(%).

## 1. Introduction

Light field imaging captures both the spatial intensity and the angular direction of light rays, providing a rich 4D representation of visual scenes (Wu et al., 2017; Wagner et al., 2021). Compared to conventional 2D images, this additional angular dimension enables the extraction of subtle geometric cues, such as disparity and occlusion boundaries. These cues are essential for achieving high-fidelity semantic segmentation in complex environments. Despite these advantages, the high dimensionality and inherent redundancy of 4D light field data pose significant computational challenges for dense prediction tasks (Zhou et al., 2021).

The evolution of light field semantic segmentation has transitioned from early CNN-based architectures (Song et al., 2019; Wu et al., 2023a) to more recent Transformer-based models (Thisanke et al., 2023; Wang et al., 2022). While CNNs struggle to capture long-range dependencies due to their intrinsic locality bias, Transformers utilize self-attention to model global contexts effectively. However, their quadratic computational complexity $O(N^2)$ creates a prohibitive bottleneck when processing high-resolution 4D data (Khan et al., 2022). This limitation often forces researchers to use aggressive downsampling, which tends to destroy fine-grained EPI structures. Since EPIs represent the fundamental geometric signature, losing this information significantly degrades performance.

This dilemma has motivated the exploration of linear-

---

[1]Engineering Research Center of Learning-Based Intelligent System, Ministry of Education, Tianjin University of Technology, Tianjin 300384, China [2]Key Laboratory of Computer Vision and System, Ministry of Education, Tianjin University of Technology, Tianjin 300384, China. Correspondence to: Chen Jia <jiachen@email.tjut.edu.cn>.

complexity alternatives. Recently, architectures like VR-WKV (Peng et al., 2023; Duan et al., 2025) and Mamba (Gu & Dao, 2024) have emerged as promising backbones, offering global receptive fields with linear efficiency $O(N)$. While successful in other domains, adapting these models to 4D light fields remains difficult. As visually evidenced in Figure 1(b), existing methods struggle to break the performance ceiling across diverse benchmarks. While they attempt to handle the data, they consistently fail to reach the high-fidelity segmentation results required for complex urban scenes, leaving a distinct accuracy gap compared to our proposed solution.

Standard linear models typically rely on naive tokenization and fixed scanning strategies, which brute-force high-dimensional manifolds into flattened 1D sequences. This serialization process disrupts the intricate of EPIs (Wu et al., 2019; Zhou et al., 2023), where linear slopes encode depth information. Consequently, these geometry-agnostic scanners fail to distinguish strictly geometric dependencies from semantic contexts. This leads to a phenomenon we term Structural Collapse, where the model falters in texture-less or occluded regions because it has traded structural integrity for computational speed.

To address this conflict, we propose $E^2$I-VRWKV, an explicit EPI-Representation and Interaction-aware network designed to bridge the gap between Geometry-aware constraints and efficient representation learning. Unlike previous approaches that treat geometry merely as an implicit feature, we elevate it to an explicit constraint within the network design. Our method incorporates an EPI Geometric Prior Generator to anchor the network to physical properties before feature encoding. Furthermore, we design the LF-ECMA block. Within this block, we introduce a GC-Gate mechanism. This module functions as a geometrically modulated aperture that dynamically calibrates the fusion of spatial and angular manifolds based on explicit disparity cues. This design ensures that the efficiency of linear models does not come at the expense of structural sensitivity.

Empirically, our method demonstrates improved capability in resolving the conflict between accuracy and cost. As highlighted in Figure 1(a), $E^2$I-VRWKV occupies the optimal position in the efficiency-performance landscape. It achieves SOTA performance while maintaining a significantly lower computational footprint compared to Transformer-based methods like OAFuser (Teng et al., 2024).

To summarize, our main contributions are as follows:

- We propose $E^2$I-VRWKV, the first Geometry-aware linear backbone tailored for light field semantic segmentation. By embedding explicit EPI priors into the scanning process, our method effectively resolves the

Structural Collapse inherent in naive linearization, enabling high-fidelity 4D parsing.

- We design the LF-ECMA block, which synergises a disparity-sensitive EPI Geometric Prior Generator with a GC-Gate mechanism. The GC-Gate acts as a geometrically modulated aperture, dynamically calibrating the fusion of spatial and angular manifolds to enforce structural consistency.

- Extensive experiments on the UrbanLF dataset (Sheng et al., 2022) demonstrate that our approach achieves a improved trade-off between accuracy and efficiency. Specifically, $E^2$I-VRWKV sets a new SOTA on UrbanLF-Real with 86.55% mIoU, significantly outperforming Transformer-based methods while maintaining a minimal computational footprint.

## 2. Related Work

### 2.1. Semantic Segmentation for Light Fields

Light field semantic segmentation entails the parsing of dense 4D spatial-angular tensors, a task uniquely challenged by the need to resolve high-dimensional geometric cues within massive 4D volumes. Early methodologies predominantly utilized CNNs to process Sub-Aperture Images (SAIs) (Liu et al., 2022) or stacked EPIs. Frameworks like LF-IENet (Cong et al., 2023) and LF-SemiNet (Zhang et al., 2024) employed specific angular convolutions to extract depth features. However, constrained by the intrinsic locality bias of convolution, these architectures often struggle to capture the long-range dependencies essential for ensuring global spatial-angular consistency. To overcome this limitation, Transformer-based methods such as OAFuser (Teng et al., 2024) and SA-Gate (Chen et al., 2020) leverage self-attention mechanisms to model global contexts. Despite their improved performance, their quadratic computational complexity $O(N^2)$ creates a prohibitive bottleneck when processing high-resolution 4D tensors. This computational burden often necessitates aggressive downsampling, which tends to destroy fine-grained EPI structures. Consequently, there is a growing need for architectures that achieve linear computational efficiency while explicitly preserving these vital geometric priors.

### 2.2. Linear Complexity Vision Backbones

The pursuit of efficiency has catalyzed the resurgence of Recurrent Neural Networks (RNNs) (Mienye et al., 2024) and the development of State Space Models (SSMs) (Liu et al., 2024; Li et al., 2024). Recently, linear attention paradigms have emerged as effective alternatives to Transformers, offering global receptive fields with linear complexity $O(N)$. Specifically, VRWKV utilizes a channel-wise time decay mechanism to enable stable, attention-free linear mixing.

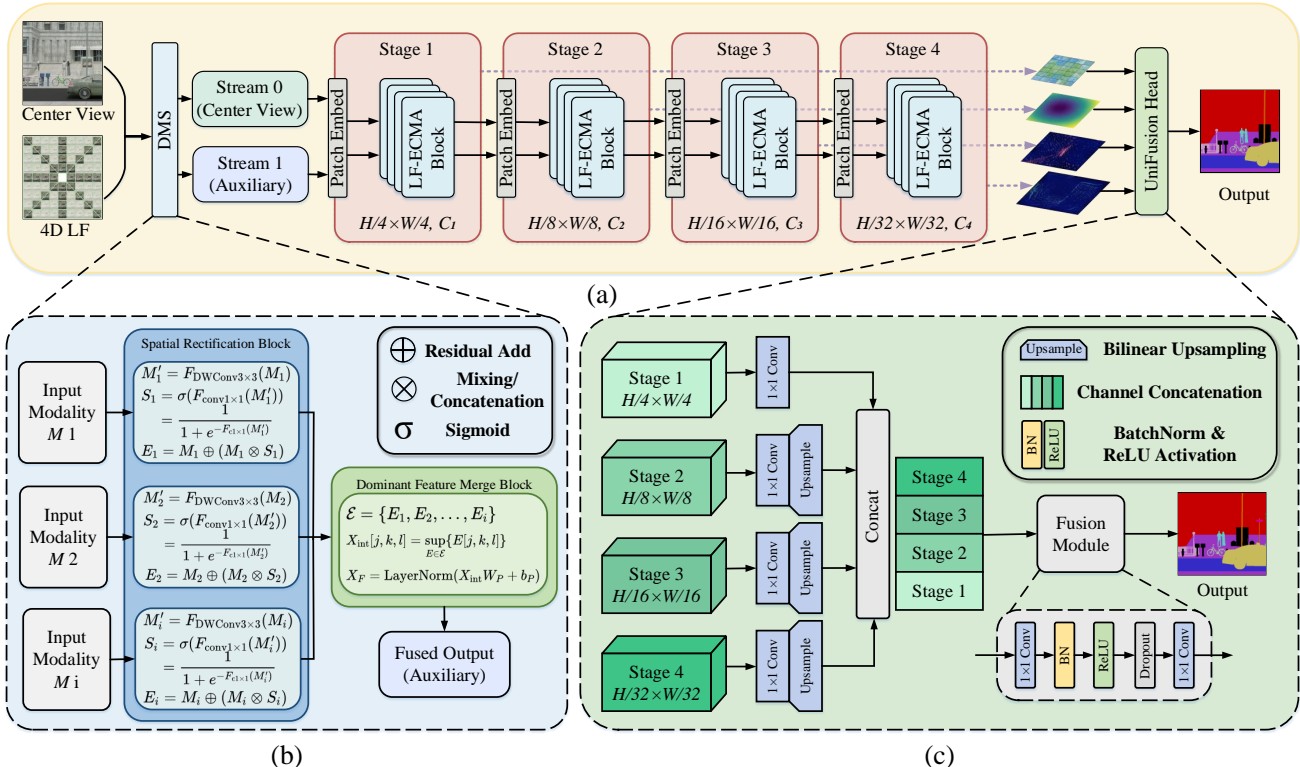

*Figure 2.* **Overall network of E²I-VRWKV**. (a) llustrates the overall architecture of E²I-VRWKV (b) displays the structure of the DMS module used for bidirectional feature rectification at each stage. (c) displays the UniFusion Head. The input undergoes comprehensive spatial-angular feature extraction, while UniFusion generates the final high-quality pixel-level prediction.

However, while these models show promise for generic 2D tasks, their application to 4D light fields faces a critical modality gap. Standard linear models rely on naive tokenization that flattens high-dimensional manifolds into 1D sequences. This process severs the local neighborhood correlations essential for disparity estimation. Consequently, purely data-driven mixers fail to capture implicit disparity cues encoded in EPI slopes. To date, adapting linear backbones to strictly respect the 4D geometric constraints of light field data remains an open challenge.

### 2.3. Geometry-Aware Interaction and Fusion

Effective LF segmentation demands the synergistic fusion of spatial texture and angular geometry (Yan et al., 2025). In light field imaging, the EPI structure acts as a strong physical prior, where lines with varying slopes correspond to different depths. Existing approaches often treat these geometric cues merely as implicit features to be learned via generic layers or combined through simple concatenation. However, such data-agnostic fusion is particularly fragile in linear models (Zhang et al., 2026; Han et al., 2025), which lack the massive parameter capacity of Transformers to implicitly recover lost geometric structures. We argue that geometry should serve as a guiding constraint rather than

a supplementary feature (Wu et al., 2023b). Unlike naive fusion methods, our proposed GC-Gate mechanism functions as a geometrically modulated aperture, dynamically calibrating the injection of angular information based on explicit EPI priors to ensure structural fidelity.

## 3. Method

### 3.1. Preliminary

We present E²I-VRWKV, a unified network designed to resolve the fundamental conflict between high-dimensional light field data representation and computational efficiency. As illustrated in Figure 2(a), the overall pipeline adopts a hierarchical Dual-Modal Stream architecture to disentangle semantic context from geometric consistency. Formally, let the input 4D light field be denoted as a tensor $\mathcal{I} \in \mathbb{R}^{U \times V \times H \times W \times C}$, where $U \times V$ represents the angular resolution and $H \times W$ denotes the spatial resolution. We explicitly decouple this manifold into two parallel feature streams: the Central View Stream $\mathbf{X}_s \in \mathbb{R}^{H \times W \times C}$ which preserves high-frequency texture details, and the EPI-Geometry Stream $\mathbf{X}_a \in \mathbb{R}^{(UV) \times H \times W \times C}$ which encodes the angular disparity cues. To facilitate deep Combination between these streams, we introduce the Dual-Modal

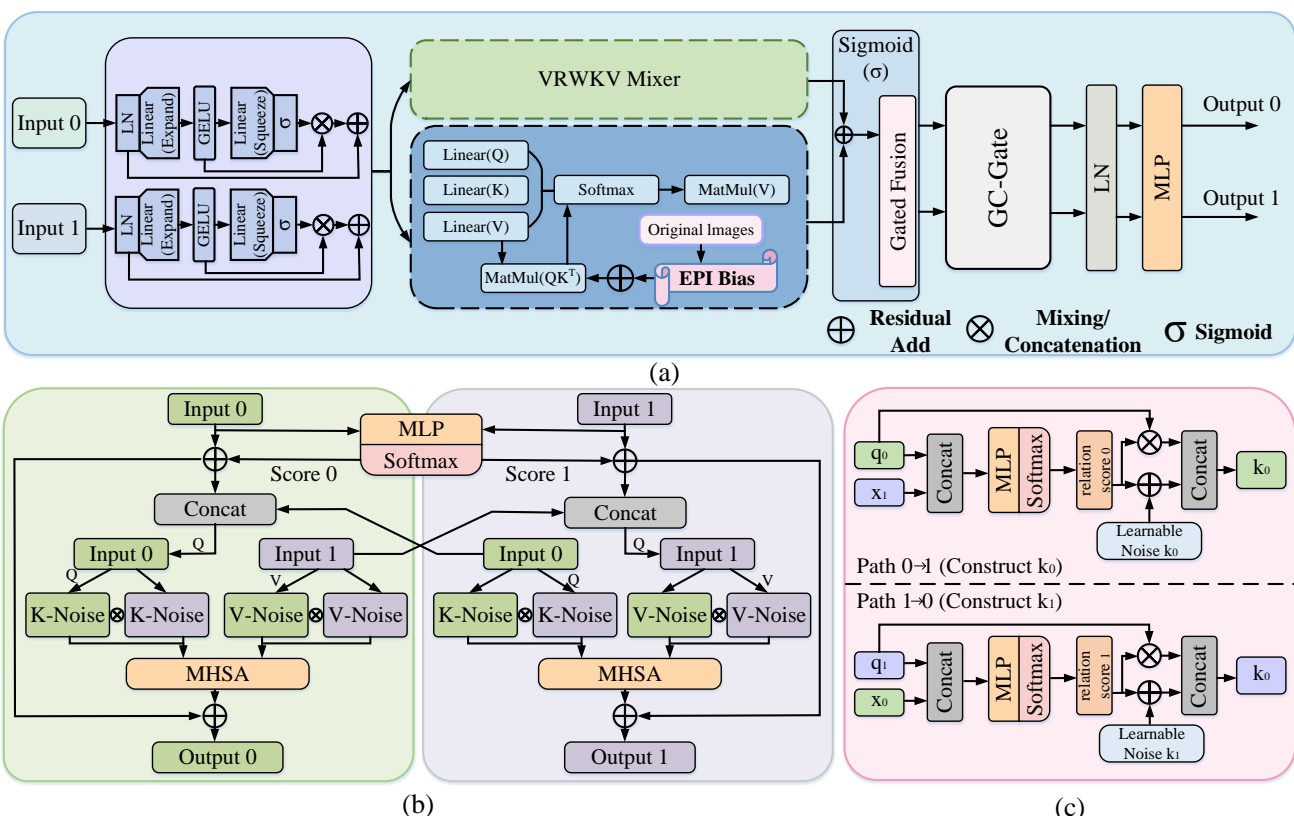

*Figure 3.* **Details of the LF-ECMA Block.** (a) illustrates the LF-ECMA architecture, which synergizes the linear-complexity Vision-RWKV Mixer for global semantics with the EPI-guided branch for structural precision. (b) details the GC-Gate mechanism. Acting as a geometrically modulated aperture, it dynamically calibrates the injection of angular cues based on the explicit EPI Prior to enforce structural consistency. (c) displays the GatedFusion module, which adaptively weights and aggregates the dual-stream features for robust representation learning.

Stream (DMS) module in Figure 2(b), which performs bidirectional feature rectification. Unlike naive fusion, DMS models the cross-modal interaction as a reciprocal residual update. Let $\mathbf{X}_s^{(l)}$ and $\mathbf{X}_a^{(l)}$ denote the features at layer $l$. The spatial-to-angular rectification is formulated as:

$$\mathcal{M}_{s \to a} = \mathcal{F}_{dw}(\mathbf{X}_s^{(l)}) \otimes \sigma(\mathcal{F}_{pw}(\mathbf{X}_s^{(l)})) \tag{1}$$

$$\mathbf{X}_a^{(l+1)} = \mathbf{X}_a^{(l)} + \eta_a \cdot \Psi(\mathcal{M}_{s \to a}) \odot \mathbf{X}_a^{(l)} \tag{2}$$

where $\mathcal{F}_{dw}$ and $\mathcal{F}_{pw}$ represent depth-wise and point-wise convolutions respectively, $\sigma$ is the Sigmoid activation, $\Psi$ denotes a spatial alignment function, and $\eta_a$ is a learnable scaling factor. Conversely, the angular-to-spatial flow injects geometric constraints into the semantic stream via:

$$\mathcal{M}_{a \to s} = \mathcal{P}_{pool}(\mathbf{X}_a^{(l)}) \tag{3}$$

$$\mathbf{X}_s^{(l+1)} = \mathbf{X}_s^{(l)} + \eta_s \cdot \text{Tanh}(\mathcal{W}_{proj}(\mathcal{M}_{a \to s})) \tag{4}$$

where $\mathcal{P}_{pool}$ represents an angular pooling operation to condense 4D information into 2D spatial maps. Following the hierarchical feature extraction, the multi-scale representations are aggregated by the UniFusion Head as depicted in

Figure 2(c). Let $\{\mathbf{F}_k\}_{k=1}^4$ be the feature maps from the four stages with strides $\{4, 8, 16, 32\}$. We project these distinct manifolds into a unified embedding space $\mathbb{R}^D$ and align their resolutions via bilinear interpolation $\mathcal{U}(\cdot)$:

$$\hat{\mathbf{F}}_k = \mathcal{U}_{H \times W}(\Phi_k(\mathbf{F}_k)), \quad \Phi_k(\mathbf{x}) = \mathbf{W}_k \mathbf{x} + \mathbf{b}_k \tag{5}$$

The final segmentation map $\mathbf{Y}_{pred}$ is derived through channel-wise concatenation followed by a dense classification head:

$$\mathbf{F}_{global} = \overset{4}{\underset{k=1}{\Big\|}} \hat{\mathbf{F}}_k \tag{6}$$

$$\mathbf{Y}_{pred} = \text{Softmax}(\mathcal{H}_{cls}(\mathbf{F}_{global})) \tag{7}$$

where $\big\|$ denotes the concatenation operator along the channel dimension.

### 3.2. EPI Geometric Prior Generator

The core challenge in light field segmentation lies in recognizing depth discontinuities within texture-sparse regions.

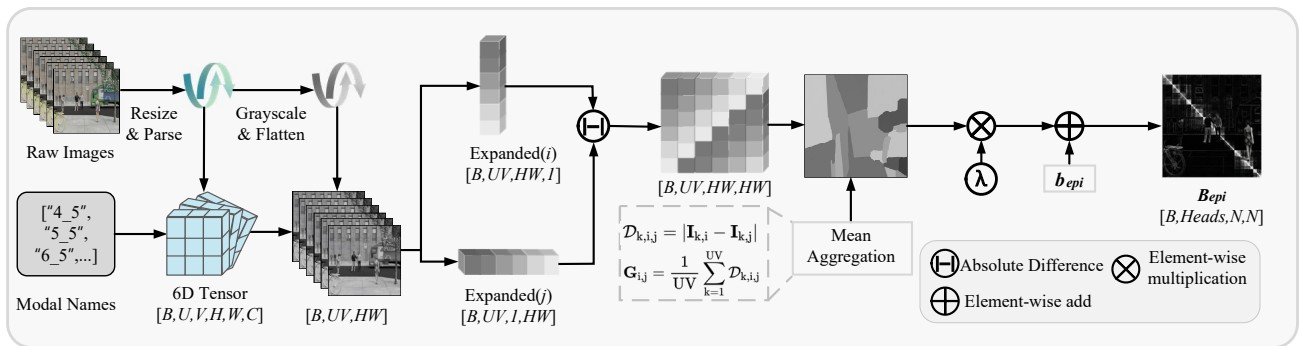

*Figure 4.* Illustration of the EPI Geometric Prior Generator.

To explicitly capture these cues, we design the EPI Geometric Prior Generator in Figure 4, which leverages the property that depth is encoded as linear slopes in EPIs. We first construct the horizontal EPI tensor $\mathbf{E}_h \in \mathbb{R}^{(U \cdot H) \times W \times C}$ by stacking angular views. A critical design choice is the utilisation of Multi-Head Self-Attention (MHSA) as the structural vessel for this prior, as it inherently calculates an $N \times N$ affinity matrix capable of modeling global geometric correlations. While MHSA exhibits quadratic complexity, we restrict its application exclusively to the EPI Prior Generator, decoupling geometric reasoning from the linear semantic backbone. We define the geometric query, key, and value projections as:

$$\mathbf{Q}_{epi} = \mathbf{E}_h \mathbf{W}_Q^e, \quad \mathbf{K}_{epi} = \mathbf{E}_h \mathbf{W}_K^e, \quad \mathbf{V}_{epi} = \mathbf{E}_h \mathbf{W}_V^e \tag{8}$$

The geometric consistency map $\mathbf{A}_{epi}$, which explicitly encodes the slope coherence between pixels, is computed with a relative positional encoding $\mathbf{P}_{rel}$ to preserve translation invariance:

$$\mathbf{A}_{epi}(i,j) = \text{Softmax}\left(\frac{(\mathbf{Q}_{epi}\mathbf{K}_{epi}^\top)_{ij} + \lambda \cdot \mathcal{G}_{ij} + \mathbf{P}_{rel}^{ij}}{\sqrt{d_k}}\right) \tag{9}$$

The aggregated geometric feature $\mathbf{G}_{epi}$ is then obtained by:

$$\mathbf{G}_{epi} = \mathbf{A}_{epi}\mathbf{V}_{epi} \tag{10}$$

Finally, we aggregate the geometric consistency map $\mathbf{A}_{epi}$ to formulate the EPI Bias $\mathcal{B}_{epi}$, serving as a learnable inductive bias tensor that will be injected into the main backbone:

$$\mathcal{B}_{epi} = \text{LN}(\gamma \cdot \text{Aggregate}(\mathbf{A}_{epi}) + b_{epi}) \tag{11}$$

where $\text{Aggregate}(\cdot)$ denotes the aggregation operation, $\gamma$ and $b_{epi}$ are learnable affine parameters, and LN represents Layer Normalization. This explicit structural modulation ensures that the network strictly enforces geometric consistency.

### 3.3. Light Field Epipolar-Aware Cross-Modal Attention

The LF-ECMA block in Figure 3(a) constitutes the computational engine of our network, designed to interplay global

semantic context with local geometric precision. It operates via two complementary branches: the Vision-RWKV Mixer for low-frequency global reasoning and the GC-Gate for high-frequency structural refinement.

#### 3.3.1. VISION-RWKV MIXER

The semantic branch utilises the Vision-RWKV backbone (Duan et al., 2025), which reformulates the recurrent update as a linear attention mechanism. For a flattened spatial sequence $\mathbf{X}_s \in \mathbb{R}^{L \times D}$, the receptance $\mathbf{r}$, key $\mathbf{k}$, and value $\mathbf{v}$ vectors at time step $t$ are derived via learned time-mixing factors $\mu$:

$$\mathbf{k}_t = \mathbf{W}_k(\mu_k \odot \mathbf{x}_t + (1 - \mu_k) \odot \mathbf{x}_{t-1}) \tag{12}$$

$$\mathbf{v}_t = \mathbf{W}_v(\mu_v \odot \mathbf{x}_t + (1 - \mu_v) \odot \mathbf{x}_{t-1}) \tag{13}$$

The core linear complexity is achieved through a channel-wise recurrence with a time-decay parameter $\mathbf{w}$. The numerator $\boldsymbol{\alpha}_t$ and denominator $\boldsymbol{\beta}_t$ states are updated recursively:

$$\boldsymbol{\alpha}_t = e^{-\mathbf{w}} \odot \boldsymbol{\alpha}_{t-1} + e^{\mathbf{k}_t} \odot \mathbf{v}_t \tag{14}$$

$$\boldsymbol{\beta}_t = e^{-\mathbf{w}} \odot \boldsymbol{\beta}_{t-1} + e^{\mathbf{k}_t} \tag{15}$$

The output of the WKV mechanism, $\mathbf{o}_{wkv}$, is computed by normalizing the accumulated states, preserving the global receptive field without quadratic cost:

$$\mathbf{o}_{wkv}^{(t)} = \sigma(\mathbf{r}_t) \odot \frac{\boldsymbol{\alpha}_{t-1} \odot e^{-\mathbf{w}+\mathbf{k}_t} + e^{\mathbf{k}_t} \odot \mathbf{v}_t}{\boldsymbol{\beta}_{t-1} \odot e^{-\mathbf{w}+\mathbf{k}_t} + e^{\mathbf{k}_t}} \tag{16}$$

#### 3.3.2. GEOMETRIC-CONTEXT GATING FUSION

To integrate the 4D angular information without disrupting the spatial manifold, we propose the GC-Gate module in Figure 3(b). Unlike naive concatenation, GC-Gate functions as a geometrically modulated aperture. It dynamically generates a gating tensor $\boldsymbol{\Gamma}$ by conditioning the angular features $\mathbf{X}_a$ on the spatial context $\mathbf{X}_s$ and the pre-computed EPI Bias $\mathcal{B}_{epi}$:

$$\boldsymbol{\Gamma} = \sigma\left(\mathcal{W}_g([\mathbf{X}_s \| \mathcal{A}_{align}(\mathbf{X}_a)]) + \lambda \cdot \mathcal{B}_{epi}\right) \tag{17}$$

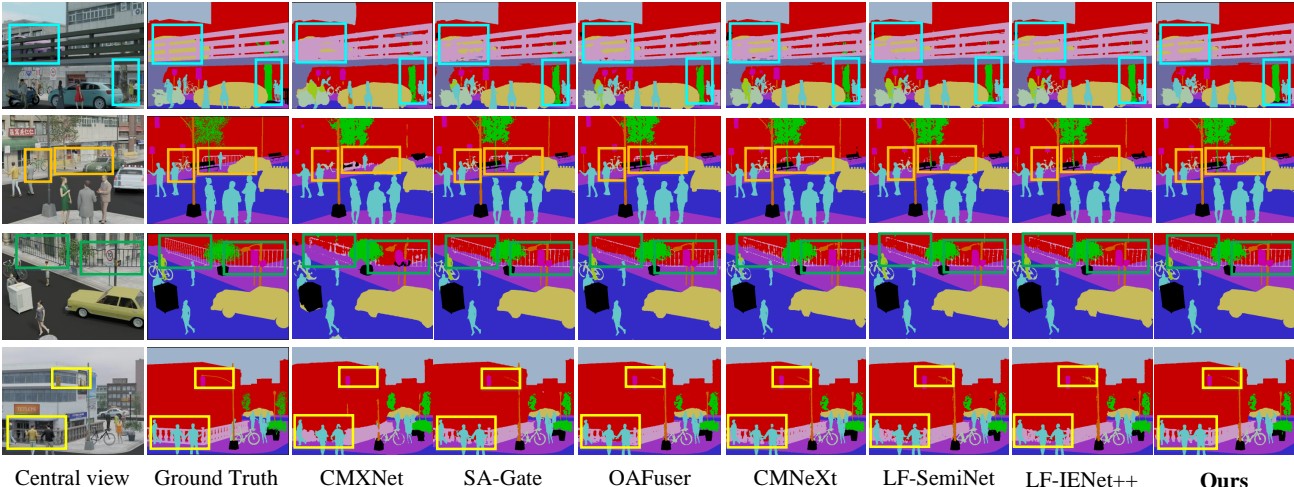

| Central view | Ground Truth | CMXNet | SA-Gate | OAFuser | CMNeXt | LF-SemiNet | LF-IENet++ | **Ours** |

*Figure 5.* The visual comparison of SOTA methods on the UrbanLF-Syn synthetic dataset.

*Table 1.* Dataset statistics of UrbanLF-Syn, UrbanLF-Real, UrbanLF-Syn-Small and UrbanLF-Syn-Big.

| Dataset | Angular Resolution | Train Set | Val Set | Test Set |
|---|---|---|---|---|
| UrbanLF-Syn | 640×480 | 172 | 28 | 50 |
| UrbanLF-Real | 623×432 | 580 | 80 | 64 |
| UrbanLF-Syn-Small | 640×480 | 280 | 40 | 80 |
| UrbanLF-Syn-Big | 640×480 | 280 | 40 | 80 |

where $\|$ represents concatenation, $\mathcal{A}_{align}$ is a spatial alignment operator, and $\lambda$ is a learnable scalar initialized to 0. The final fused feature $\mathbf{F}_{out}$ is synthesized by the GatedFusion mechanism in Figure 3(c), which adaptively balances the global semantic stream and the geometry-guided stream:

$$\mathbf{F}_{out} = \mathbf{X}_s + \mathbf{\Gamma} \odot (\mathcal{W}_{fuse}(\mathbf{X}_a) \otimes \mathcal{B}_{epi}) \quad (18)$$

This formulation ensures that angular information is only injected when it is geometrically consistent with the spatial structure, effectively resolving occlusion ambiguities.

# 4. Experiments

## 4.1. Datasets

We perform comprehensive evaluations on the UrbanLF dataset (Sheng et al., 2022). UrbanLF-Real consists of 724 samples captured by a Lytro Illum camera, featuring complex urban lighting and severe occlusions. UrbanLF-Syn provides 250 synthetic samples with precise ground truth to benchmark theoretical upper bounds. To further probe the model's sensitivity to disparity magnitude, it includes specific splits of Syn-Small and Syn-Big, allowing for a granular analysis of performance across varying geometric complexities.

## 4.2. Implementation Details

**Experimental Settings.** We implemented Our network using PyTorch v1.12.1 and trained it on an Intel Xeon Platinum 8370C CPU paired with 8 NVIDIA GeForce RTX 4090 GPUs. Input images are randomly cropped to $640 \times 480$. We utilise the AdamW optimizer with an initial learning rate of $6 \times 10^{-5}$ and a weight decay of $0.01$. The learning rate follows a polynomial decay schedule with a power of 0.9 over 500 epochs. Standard data augmentation techniques are employed to prevent overfitting. Crucially, all Light Field baseline methods consumed the exact same 4D LF inputs as our model, while conventional 2D baselines were fed the corresponding 2D center views. All models were evaluated under strictly identical resolutions and training augmentations to ensure fair comparison.

**Comparison Methods.** To comprehensively evaluate our model, we compared E²I-VRWKV with 20 SOTA methods, including IF-SemiNet (Zhang et al., 2024), MSIGNet (Li et al., 2023), OCRNet (Yuan et al., 2020), SA-Gate (Chen et al., 2020), CMXNet (Zhang et al., 2023a), Mask2Former (Cheng et al., 2022), SegFormer (Xie et al., 2021), GeminiFusion (Jia et al., 2024), LF-IENet (Cong et al., 2023), LF-IENet++ (Cong et al., 2024), MMS-Forer (Reza et al., 2024), CMNeXt (Zhang et al., 2023b), OAFuser (Teng et al., 2024), EPI-Mamba (Li et al., 2025), DAVSS (Zhuang et al., 2020), TMANet (Wang et al., 2021), PSPNet (Zhao et al., 2017), SETR (Zheng et al., 2021), MTINet (Vandenhende et al., 2020), ESANet (Seichter et al., 2021) and OCR-LF (Sheng et al., 2022).

**Evaluation Metrics.** We evaluate performance using Mean Accuracy (mAcc), and Mean Intersection over Union (mIoU). We also report parameters and GFLOPs. The mIoU

*Table 2.* Comparison with SOTA methods on UrbanLF-Syn and UrbanLF-Real datasets. The best results are highlighted in green and the second best are blue .

| Method | Backbone | Params(M) | UrbanLF-Syn | | | UrbanLF-Real | | |
|--------|----------|-----------|------|------|------|------|------|------|
| | | | Acc | mAcc | mIoU | Acc | mAcc | mIoU |
| LF-SemiNet (Zhang et al., 2024) | HRNetV2-W48 | 80.1 | 89.42 | 83.04 | 73.36 | 91.41 | 82.47 | 76.50 |
| MSIGNet (Li et al., 2023) | HrNetV2-W48 | 76.2 | 92.00 | 87.71 | 80.33 | 91.12 | 83.91 | 76.65 |
| OCRNet (Yuan et al., 2020) | HRNetV2-W48 | 70.5 | 92.58 | 87.21 | 80.51 | 92.55 | 85.40 | 79.14 |
| SA-Gate (Chen et al., 2020) | ResNet-101 | 219.0 | 91.36 | 86.17 | 78.20 | 92.30 | 86.29 | 79.33 |
| CMXNet (Zhang et al., 2023a) | MiT-B5 | 181.1 | 92.58 | 86.66 | 77.57 | 91.03 | 85.29 | 79.57 |
| Mask2Former (Cheng et al., 2022) | ResNet-101 | **43.95** | 91.24 | 86.47 | 79.61 | 91.43 | 85.75 | 78.33 |
| SegFormer (Xie et al., 2021) | MiT-B5 | 82.0 | 92.51 | 87.67 | 80.54 | 93.48 | 87.12 | 81.37 |
| GeminiFusion (Jia et al., 2024) | MiT-B4 | 103.0 | 92.04 | 87.41 | 80.75 | 93.00 | 88.41 | 82.16 |
| LF-IENet (Cong et al., 2023) | HRNetV2-W48 | 117.4 | 93.04 | 88.18 | 81.22 | 93.74 | 88.12 | 82.09 |
| LF-IENet++ (Cong et al., 2024) | HRNetV2-W48 | 124.7 | 92.41 | 88.31 | 81.78 | 93.02 | 86.69 | 80.35 |
| MMSFormer (Reza et al., 2024) | MiT-B4 | 173.0 | 91.24 | 87.57 | 81.79 | 92.54 | 88.64 | 83.07 |
| OAFuser (Teng et al., 2024) | MiT-B4 | 164.1 | 93.42 | 88.22 | 81.93 | 94.08 | 87.74 | 82.21 |
| EPI-Mamba (Li et al., 2025) | VMamba-S | 81.8 | 94.06 | 89.86 | 84.43 | 94.09 | 89.66 | 84.07 |
| EPI-Mamba (Li et al., 2025) | VMamba-B | 120.8 | 94.16 | 90.61 | 84.99 | 94.19 | 90.09 | 84.13 |
| **Ours** | MiT-B4 | 103.0 | **95.53** | **92.09** | **86.46** | **95.12** | **92.59** | **86.55** |

is calculated as:

$$\text{mIoU} = \frac{1}{K} \sum_{i=0}^{K} \frac{TP_i}{TP_i + FP_i + FN_i} \quad (19)$$

where $K$ denotes the number of classes.

### 4.3. Comparison with SOTA Methods

As listed in Table 2, compared with 13 other SOTA methods, our proposed $E^2I$-VRWKV consistently achieves the best performance across both UrbanLF-Real and UrbanLF-Syn datasets.

On the UrbanLF-Real dataset, $E^2I$-VRWKV demonstrates considerable robustness, achieving the highest performance among all competing methods. Notably, it surpasses the next best method by a substantial margin of 4.62% in mIoU. Existing multi-modal fusion frameworks, such as CMX and MMSFormer, typically treat auxiliary angular views merely as supplementary feature channels, performing fusion primarily in the channel dimension. Our significant improvement is directly attributed to the robust ability of the EPI Geometric Prior Generator to capture intrinsic depth cues even in large occlusion areas. By internalizing the EPI structure into the feature learning process, our model effectively distinguishes foreground objects from cluttered backgrounds, significantly enhancing representational power in unconstrained real-world scenarios.

On the UrbanLF-Syn dataset, our method surpasses all other SOTA methods in every metric, outperforming the leading light field method OAFuser by 1.85% in mIoU. The improved performance of $E^2I$-VRWKV demonstrates the effectiveness of the GC-Gate mechanism and the Interaction-Aware VRWKV in capturing fine textures and elongated

*Table 3.* Computational efficiency and inference latency comparison. The best results are highlighted in green and the second best are blue .

| Method | Params (M) | GFLOPs | Latency (ms) | mIoU (%) |
|--------|-----------|--------|--------------|----------|
| MMSFormer | 173.0 | 160.0 | 86.4 | 81.79 |
| OAFuser | 164.1 | 188.6 | 52.4 | 81.93 |
| EPI-Mamba | 120.8 | N/A | N/A | 84.99 |
| **Ours** | **103.0** | **221.0** | **44.1** | **86.46** |

structures. Unlike standard attention mechanisms that may blur high-frequency details, our approach preserves the integrity of thin structures by enforcing geometric consistency along the epipolar lines.

We further provide a granular analysis of category-specific performance as detailed in Table 4 and Table 5. Our method exhibits dominant advantages in geometrically complex categories. Specifically, for thin and slender categories such as Bridge and Building, $E^2I$-VRWKV outperforms the second-best approach by clear margins of 13.5% and 17.3%, respectively. These categories are inherently challenging for semantic segmentation due to their narrow structures and texture similarity with the background. The distinct performance leap in these classes verifies that our Explicit EPI Prior successfully guides the network to rely on robust geometric scaffoldings rather than superficial texture correlations, effectively preventing the structural collapse often observed in other methods.

As illustrated in Figure 5, our method produces clearer and more precise segmentation maps with improved detail capture in typical scenarios compared to other methods. In challenging regions containing thin objects like bicycle spokes, street poles, and distant pedestrians, standard methods tend

*Table 4.* The quantitative comparison results of mainstream SOTA methods based on LF and RGBD on the UrbanLF-Syn dataset. Best results are in green , and second best results are blue .

| Method | Bike | Building | Fence | Others | Person | Pole | Road | Sidewalk | T.Sign | Veg. | Vehicle | Bridge | Rider | Sky | mIoU | mAcc | Acc |
|---|---|---|---|---|---|---|---|---|---|---|---|---|---|---|---|---|---|
| CMXNet | 73.54 | 65.08 | 87.52 | 69.61 | 85.19 | 76.53 | 91.82 | 91.67 | 77.93 | 68.96 | 91.33 | 64.52 | 54.05 | 94.03 | 77.57 | 86.66 | 92.58 |
| SA-Gate | 75.13 | 63.95 | 89.63 | 69.80 | 86.71 | 75.77 | 94.62 | 91.15 | 76.03 | 67.70 | 92.51 | 66.13 | 55.77 | 90.68 | 78.20 | 86.17 | 91.36 |
| MTINet | 73.61 | 64.43 | 87.20 | 68.79 | 84.16 | 72.21 | 90.91 | 89.34 | 74.08 | 64.43 | 91.63 | 66.36 | 52.11 | 93.14 | 76.60 | 84.45 | 90.27 |
| LF-SemiNet | 72.58 | 69.38 | 77.37 | 65.82 | 78.28 | 74.94 | 81.87 | 79.61 | 77.41 | 60.73 | 78.41 | 62.34 | 65.85 | 82.36 | 73.36 | 83.04 | 89.42 |
| LF-IENet | 75.65 | 72.20 | 92.07 | 70.17 | 89.37 | 77.56 | 95.99 | 92.36 | 90.22 | 68.11 | 91.11 | 63.78 | 69.44 | 97.32 | 81.11 | 88.21 | 92.91 |
| LF-IENet++ | 76.34 | 72.81 | 92.77 | 70.83 | 90.06 | 78.18 | 96.01 | 92.78 | 90.67 | 68.61 | 91.46 | 64.02 | 70.14 | 97.97 | 82.39 | 88.67 | 93.55 |
| OAFuser | 79.38 | 70.54 | 91.66 | 73.81 | 91.65 | 85.01 | 90.03 | 93.32 | 83.39 | 72.53 | 96.83 | 70.93 | 63.95 | 95.72 | 82.69 | 88.21 | 82.69 |
| **Ours** | 77.52 | 90.11 | 77.19 | 86.10 | 90.14 | 78.01 | 96.26 | 93.60 | 92.06 | 69.78 | 95.97 | 84.43 | 81.51 | 97.72 | 86.46 | 92.09 | 95.53 |

*Table 5.* The Quantitative Comparison Results of Mainstream SOTA Methods Based on LF and RGBD on the UrbanLF-Real Dataset. Best results are in green , and second best results are blue .

| Method | Bike | Building | Fence | Others | Person | Pole | Road | Sidewalk | T.Sign | Veg. | Vehicle | Bridge | Rider | Sky | mIoU | mAcc | Acc |
|---|---|---|---|---|---|---|---|---|---|---|---|---|---|---|---|---|---|
| CMXNet | 84.41 | 91.56 | 84.84 | 45.97 | 88.03 | 72.74 | 87.88 | 66.89 | 83.18 | 84.04 | 95.03 | 85.80 | 57.05 | 87.55 | 79.57 | 85.29 | 91.03 |
| SA-Gate | 83.75 | 88.02 | 80.94 | 46.62 | 87.29 | 76.15 | 86.97 | 64.55 | 80.33 | 84.47 | 93.35 | 79.31 | 65.17 | 93.68 | 79.33 | 86.29 | 92.30 |
| MTINet | 83.93 | 89.88 | 83.36 | 44.25 | 86.57 | 71.18 | 88.43 | 64.29 | 81.69 | 83.56 | 94.61 | 85.33 | 56.40 | 93.37 | 79.06 | 86.07 | 92.14 |
| LF-SemiNet | 81.40 | 84.50 | 83.22 | 44.82 | 83.22 | 70.73 | 83.61 | 65.25 | 77.92 | 77.86 | 88.28 | 79.31 | 62.27 | 88.67 | 76.50 | 82.47 | 91.41 |
| LF-IENet | 85.90 | 91.98 | 85.05 | 45.08 | 88.63 | 72.77 | 90.56 | 65.83 | 83.23 | 85.05 | 96.43 | 86.67 | 57.62 | 95.67 | 80.49 | 86.87 | 92.82 |
| LF-IENet++ | 83.95 | 90.78 | 84.54 | 41.89 | 90.73 | 75.30 | 88.73 | 65.76 | 80.29 | 83.40 | 95.67 | 83.95 | 69.77 | 95.05 | 80.85 | 87.01 | 92.94 |
| OAFuser | 87.92 | 91.95 | 87.64 | 48.56 | 93.63 | 77.55 | 91.74 | 71.92 | 87.43 | 89.04 | 97.05 | 88.89 | 79.03 | 96.61 | 84.93 | 89.84 | 94.61 |
| **Ours** | 87.00 | 92.23 | 89.60 | 67.41 | 92.68 | 77.88 | 92.41 | 74.73 | 87.91 | 91.67 | 97.13 | 81.75 | 83.12 | 96.12 | 86.55 | 92.59 | 95.12 |

*Table 6.* Quantitative analysis of structural collapse on the UrbanLF-Syn dataset using the BT mIoU metric.

| Method | Architecture / Strategy | mIoU (%) | BT mIoU (%) |
|---|---|---|---|
| LF-IENet | CNN-based | 81.11 | 76.80 |
| LF-IENet++ | CNN-based | 82.39 | 77.30 |
| OAFuser | Transformer-based | 82.69 | 79.68 |
| Baseline | Flattened Linear | 80.20 | 66.52 |
| **Ours** | **Geometry-Aware** | **86.46** | **83.01** |

*Table 7.* Ablation Study of the Token Shift Strategy(q_shift).

| Token Shift | mIoU(%) | Acc(%) | mAcc(%) | Params(M) | FLOPs(G) |
|---|---|---|---|---|---|
| q_shift | 86.46 | 95.53 | 92.09 | 103.0 | 221.0 |
| double_shift | 85.78 | 94.02 | 91.39 | 102.9 | 218.4 |
| single_shift | 84.69 | 93.04 | 89.75 | 102.8 | 214.3 |

to produce fragmented or over-smoothed predictions. In contrast, $E^2I$-VRWKV generates sharp, continuous boundaries that closely match the ground truth. This stability verifies that the LF-ECMA block effectively anchors linear scanning to physical depth cues, ensuring consistent segmentation accuracy regardless of disparity magnitude.

## 4.4. Complexity Analysis

To offer a more comprehensive perspective on efficiency, we evaluate the inference latency of our method alongside strong Transformer-based and SSM-based baselines. The measurements are conducted on a single RTX 4090 GPU at a resolution of $640 \times 480$. As shown in Table 3, $E^2I$-VRWKV strikes a superior balance between linear scalability and geometric precision. Our approach achieves the highest mIoU while maintaining the fastest inference speed of 44.1 ms, which is a substantial improvement over the 52.4 ms required by OAFuser and the 86.4 ms needed by

MMSFormer. Furthermore, with a parameter count of only 103.0M, it utilizes a much smaller footprint compared to existing multi-modal fusion frameworks.

*Table 8.* Ablation study of the $E^2I$-VRWKV network.

| Structure | #Params(M) | GFLOPs | mIoU(%) |
|---|---|---|---|
| $E^2I$-VRWKV | 103.0 | 221.0 | 86.46 |
| – without GC-Gate | 94.5 | 198.4 | 84.68 (-1.78) |
| – without VRWKV | 99.4 | 206.3 | 83.37 (-3.09) |
| – with EPI Bias instead MHSA | 101.4 | 208.4 | 83.48 (-2.98) |
| – without VRWKV & EPI Bias | 94.5 | 192.8 | 80.20 (-6.26) |
| – without DMS | 102.1 | 219.4 | 85.32 (-1.14) |
| – without Gated Fusion | 100.3 | 215.0 | 84.98 (-1.48) |
| – with MiT-B2 | 42.3 | 110.0 | 78.02 (-8.44) |

*Table 9.* Performance analysis of our model with different MiT backbones on the UrbanLF dataset.

| Backbone | mAcc(%) | Params(M) | FLOPs(G) | mIoU(%) |
|---|---|---|---|---|
| MiT-B1 | 79.11 | 23.4 | 77.8 | 70.61(-15.85) |
| MiT-B2 | 86.10 | 42.2 | 110.0 | 78.02(-8.44) |
| MiT-B3 | 90.26 | 75.7 | 166.2 | 83.08(-3.38) |
| MiT-B4 | 92.09 | 103.0 | 221.3 | 86.46 |

## 4.5. Ablation Studies

We conducted comprehensive ablation experiments on the UrbanLF-Syn dataset to verify the effectiveness of each component in our proposed $E^2I$-VRWKV.

**Quantitative Analysis of Structural Collapse.** Naive 1D sequence flattening fundamentally disrupts the intrinsic epipolar geometry of light fields. This disruption causes a severe degradation in segmentation accuracy and boundary integrity for geometrically complex or thin structures, a phenomenon we formalize as Structural Collapse. To rigorously quantify this degradation, we introduce the Boundary

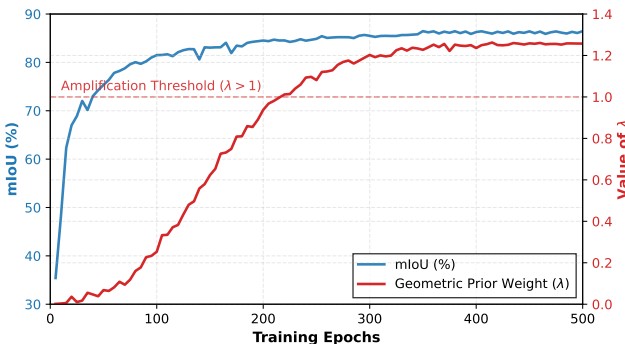

*Figure 6.* Evolution dynamics of the learnable scaling factor $\lambda$ during training. The steady growth and convergence of $\lambda$ above the amplification threshold (1.0) mathematically confirm that $E^2I$-VRWKV actively relies on explicit geometric priors rather than suppressing them as noise.

and Thin Structure mIoU (BT mIoU) metric. This metric calculates the average mIoU exclusively for strongly geometry-reliant categories, namely Pole, Bike, Bridge, and Traffic Sign. As presented in Table 6, the baseline utilizing purely flattened linear tokenization yields a BT mIoU of only 66.52%, empirically confirming the severity of this structural collapse. While existing multi-modal SOTA methods such as LF-IENet++ and OAFuser achieve improved BT mIoU scores of 77.30% and 79.68% respectively, they depend heavily on computationally expensive CNN or Transformer architectures. In sharp contrast, our explicit geometry-aware tokenization significantly elevates the BT mIoU to 83.01%, successfully outperforming all leading methods while strictly maintaining linear complexity. These results provide definitive evidence that explicit geometric injection is indispensable for preserving structural integrity against serialization loss.

**Ablation study of core components.** As listed in Table 8, we systematically analyze the contribution of the GC-Gate mechanism and the EPI Geometric Prior. When fully integrating these modules, our model achieves the best results across all metrics. Notably, replacing the geometric gating with naive concatenation leads to a performance drop of 1.78% in mIoU. This confirms that dynamic feature calibration is essential; simply mixing modalities without geometric guidance introduces noise. Similarly, removing the explicit EPI Prior results in a significant 2.98% decline in mIoU. This highlights the module's strength in capturing the anisotropic nature of 4D light fields, validating that the standard alone cannot effectively distinguish background noise from depth-consistent regions.

**Evolution of geometric consistency.** To explicitly address whether the network genuinely relies on the EPI geometric prior or merely suppresses it as noise, we trace the evolution dynamics of the learnable scaling factor $\lambda$ within the GC-Gate mechanism. As illustrated in Figure 6, the trajectory of

$\lambda$ provides definitive mathematical evidence of its structural importance. If the network were to treat the explicit EPI bias as redundant noise, the optimization process would inevitably suppress $\lambda$ toward zero. Conversely, $\lambda$ steadily climbs throughout training, ultimately converging firmly above 1.25. Concurrently, the validation mIoU exhibits a strong positive correlation with the growth of this geometric weight. This synchronous evolution mathematically proves that $E^2I$-VRWKV does not disregard the geometric bias; instead, it actively enforces the EPI representation as an indispensable physical hard constraint, thereby effectively improving the model's segmentation accuracy on complex structures.

**Ablation study of scanning strategies.** As listed in Table 7, we evaluate the necessity of multi-directional spatial modeling by varying the scan directions in the VRWKV backbone. The results demonstrate that reducing scan directions leads to a consistent performance degradation. Specifically, switching from the default quad-directional scanning to a single-directional configuration causes a severe decline of 1.94% in mIoU. This indicates that multi-directional scanning is crucial for the VRWKV mixer to preserve the topological integrity of EPI structures against serialization loss, thereby ensuring a robust global receptive field.

**Impact of backbone scale.** To investigate the scalability of our approach, we evaluate performance across various backbone capacities from MiT-B1 to MiT-B4. As illustrated in Table 9, there is a clear upward trend in metrics as capacity increases. Notably, MiT-B4 yields a substantial gain of 3.38% in mIoU over MiT-B3. Although this introduces a larger parameter footprint, it remains within a reasonable range for high-fidelity tasks. Considering the critical requirement for segmentation accuracy in complex urban scenes, MiT-B4 strikes an optimal balance between performance and resource efficiency.

## 5. Conclusion

In this paper, we presented E$^2$I-VRWKV, a Geometry-aware linear backbone that efficiently resolves the conflict between global context modeling and local geometric preservation in light field semantic segmentation. By synergizing an explicit EPI Geometric Prior with the linear-complexity VRWKV backbone via the novel LF-ECMA block, our method dynamically calibrates cross-modal interactions through the GC-Gate mechanism, effectively preventing structural collapse in texture-less or occluded regions. Extensive experiments on the UrbanLF benchmark demonstrate that E$^2$I-VRWKV establishes a new SOTA, achieving 86.55% mIoU on UrbanLF-Real.

## Acknowledgements

This work was supported by the National Natural Science Foundation of China (NSFC) under Grants T2422015 and 62306212; the Beijing-Tianjin-Hebei Natural Science Foundation Cooperation Project under Grant25JJJJC0009; the China Postdoctoral Science Foundation under Grants 2024M762376; and the Marie Skladowska-Curie Actions (MSCA) under Project No. 10111188.

## Impact Statement

This paper aims to advance autonomous urban perception by resolving the conflict between computational efficiency and geometric structural integrity in 4D light field data, thereby contributing to the safety and reliability of autonomous driving systems and robotics in complex environments. There may be some potential societal consequences of our work, none of which we feel must be specifically highlighted here.

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

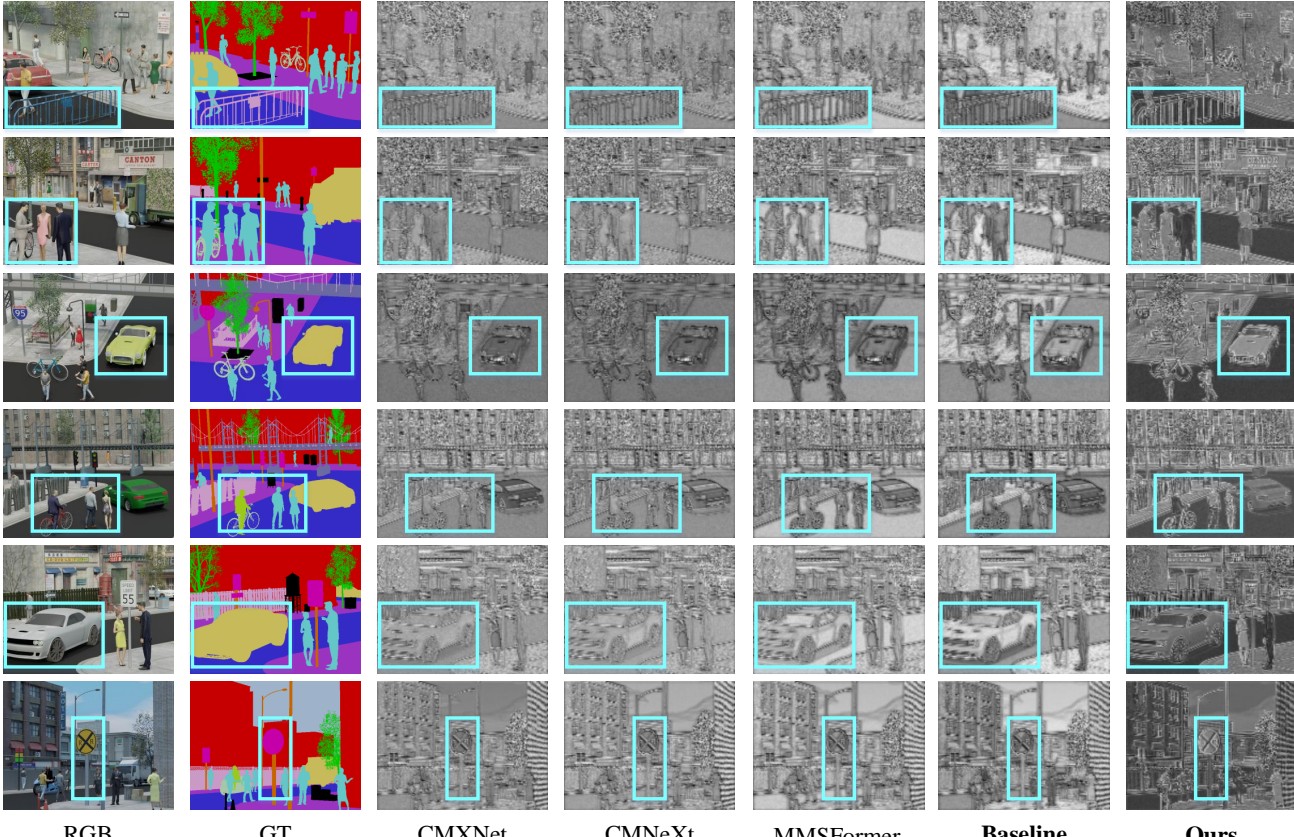

RGB       GT       CMXNet       CMNeXt       MMSFormer       **Baseline**       **Ours**

*Figure 7.* **Visualization of intermediate feature representations.** We visualize the feature maps from the first stage to illustrate the internal geometric perception capability. The Baseline exhibits activation patterns primarily dominated by visual textures. In contrast, our method effectively suppresses irrelevant background noise and emphasizes structural boundaries, providing a clear geometric basis that leads to improved segmentation accuracy compared to SOTA methods. Blue boxes highlight critical details.

## A. Visualisation Comparisons.

To visually demonstrate the advantages of $E^2I$-VRWKV, we present detailed visual results in Figure 7, including the final semantic segmentation maps and intermediate feature maps extracted from the block of Stage 1. For the UrbanLF-Syn dataset (Sheng et al., 2022), which primarily characterizes complex urban scenarios with severe occlusions, varying disparities, and intricate object boundaries, our method consistently achieves accurate segmentation, capturing even intricate geometric details. This is attributed to the strong capability of the EPI Geometric Prior Generator in capturing the underlying angular-spatial morphology, which effectively guides the network to distinguish depth discontinuities from visual textures at an early stage.

In comparison with SOTA methods like CMX (Zhang et al., 2023a), CMNeXt (Zhang et al., 2023b), and MMSFormer (Reza et al., 2024), our approach demonstrates significant superiority in boundary continuity and fine-grained structure segmentation. Specifically, while MMSFormer achieves clear results by leveraging Transformer-based cross-attention, it treats the auxiliary angular views merely as distinct modalities, similar to generic RGB-D fusion. Consequently, it overlooks the intrinsic epipolar geometry embedded in the light field structure and fails to enforce strict geometric consistency along the epipolar lines. This limitation leads to the structural collapse of fine-grained boundaries in its predictions. For instance, when handling thin objects such as the bicycle wheels and poles shown in Figure 7, MMSFormer tends to produce disconnected or over-smoothed predictions that fail to align with the actual geometric edges, whereas our method preserves these structures by explicitly modeling the EPI cues.

Furthermore, to provide deeper insight into the internal geometric perception mechanism, we compare the intermediate feature representations of the Baseline, which is devoid of the Explicit EPI Prior and VRWKV modules, and our method. By observing the feature maps extracted from the block of Stage 1, we find that the Baseline exhibits distinct texture-dominated

*Table 10.* Comparison with SOTA methods on UrbanLF-Syn-Small and UrbanLF-Syn-Big datasets. The best results are highlighted in green and the second best are blue .

| Method | Backbone | Params(M) | UrbanLF-Small | | | UrbanLF-Big | | |
|---|---|---|---|---|---|---|---|---|
| | | | Acc | mAcc | mIoU | Acc | mAcc | mIoU |
| DAVSS (Zhuang et al., 2020) | Xception-65 | 56.0 | 88.82 | 82.49 | 73.85 | 88.70 | 82.26 | 73.62 |
| TMANet (Wang et al., 2021) | ResNet-50 | **33.4** | 88.85 | 82.09 | 74.06 | 88.98 | 81.75 | 74.11 |
| PSPNet (Zhao et al., 2017) | ResNet-101 | 67.8 | 88.91 | 83.34 | 74.59 | 88.91 | 83.34 | 74.59 |
| SETR (Zheng et al., 2021) | Vit-Base | 97.0 | 89.72 | 84.12 | 76.10 | 89.72 | 84.12 | 76.10 |
| MTINet (Vandenhende et al., 2020) | HRNetV2-W48 | 98.7 | 90.27 | 84.45 | 76.60 | 90.27 | 84.45 | 76.60 |
| OCRNet (Yuan et al., 2020) | HRNetV2-W48 | 70.4 | 90.68 | 85.30 | 77.92 | 90.68 | 85.30 | 77.92 |
| ESANet (Seichter et al., 2021) | ResNet-34 | 46.9 | 91.85 | 85.58 | 78.07 | 91.85 | 85.58 | 78.07 |
| SA-Gate (Chen et al., 2020) | ResNet-101 | 110.9 | 91.36 | 86.17 | 78.20 | 91.36 | 86.17 | 78.20 |
| OCR-LF (Sheng et al., 2022) | HRNetV2-W48 | 137.4 | 90.78 | 85.31 | 78.26 | 91.05 | 85.62 | 78.87 |
| LF-IENet (Cong et al., 2023) | HRNetV2-W48 | 117.4 | 91.30 | 86.17 | 79.23 | 91.23 | 85.40 | 79.08 |
| LF-IENet++ (Cong et al., 2024) | HRNetV2-W48 | 124.7 | 91.36 | 85.83 | 79.59 | 91.47 | 86.14 | 79.88 |
| **Ours** | MiT-B4 | 103.0 | **94.76** | **90.43** | **84.33** | **92.08** | **91.28** | **85.16** |

activations. For instance, in texture-rich regions such as road surfaces and walls, the Baseline generates excessive high-frequency noise responses, failing to clearly delineate semantic boundaries. In sharp contrast, with the injection of the explicit EPI bias, the Stage-1 feature maps of $E^2I$-VRWKV exhibit sharp and edge-aware activation patterns. This early geometric injection effectively suppresses irrelevant background texture interference and highlights key depth discontinuities, including the contours of pedestrians and vehicles, thereby establishing a clear structural framework for subsequent deep semantic modeling.

## B. Comparison with SOTA Methods.

While our main paper demonstrates domain generalization on real-world data, the precise control offered by synthetic environments allows us to rigorously test the model's adaptability to varying geometric configurations. To this end, we conduct a granular analysis on UrbanLF-Syn-Small and UrbanLF-Syn-Big. This experimental design allows us to disentangle the model's performance in two contrasting geometric regimes: sensitivity to subtle depth cues and robustness against large displacements.

On the UrbanLF-Syn-Small dataset, geometric signals are inherently weak and easily overwhelmed by texture noise, posing a significant challenge for feature extraction. As shown in Table 10, our method achieves 84.33% mIoU on UrbanLF-Syn-Small, significantly outperforming the second-best SA-Gate (Chen et al., 2020). This superiority indicates that our GC-Gate mechanism functions effectively as a geometric amplifier, successfully isolating and enhancing minute high-frequency depth signals from the background to prevent the vanishing geometry problem common in standard backbones.

Conversely, UrbanLF-Syn-Big represents the other extreme, characterized by large pixel displacements and severe occlusions that disrupt feature continuity. In this scenario, our method demonstrates significant robustness, achieving 85.16% mIoU and surpassing large-scale Transformer-based methods such as SETR (Zheng et al., 2021) and Mask2Former (Cheng et al., 2022). We attribute this stability to our Multi-directional Scanning Strategy. Unlike fixed-grid pooling or standard window-based attention which may fail to capture steep EPI slopes caused by wide baselines, our scanning approach ensures that long-range geometric dependencies are preserved along at least one scanning direction, maintaining structural integrity even when object features undergo significant spatial displacement.

## C. Feature Distribution Analysis.

To provide deeper insight into how the explicit EPI prior reshapes the internal feature space, we visualize the activation statistics of the Stage-2 Layer Normalization (LN) layers in Figure 8. Specifically, the Yellow curve represents the distribution of the MiT-B4 backbone, serving as a reference for the 2D inductive bias. The Grey curve depicts the Baseline, which is devoid of the EPI Geometric Prior and VRWKV modules, while the Blue curve represents our full $E^2I$-VRWKV.

As observed in the figure 8, the feature distribution of the Baseline exhibits a high degree of overlap with the ImageNet-pretrained prior, retaining a narrow and zero-centered Gaussian shape. This phenomenon indicates a Conservative Shift, suggesting that without explicit geometric guidance, the linear backbone struggles to break away from the 2D patterns

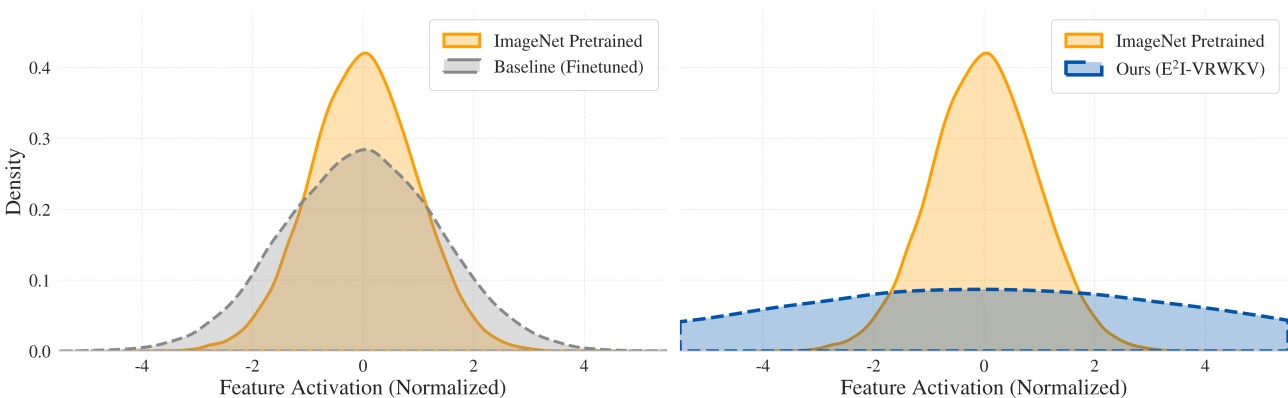

*Figure 8.* **Visualization of feature distribution in Stage-2 Layer Normalization layers.** The Yellow curve represents the MiT-B4 backbone. Left: The Baseline (Grey) exhibits a distribution highly consistent with the pretrained prior, indicating that it largely retains the 2D inductive bias without significant adaptation. Right: In contrast, our $E^2I$-VRWKV (Blue) demonstrates a significantly flattened distribution with a wider dynamic range. This variance shift suggests that the introduced EPI geometric priors drive the backbone to reconstruct its feature space to accommodate the complex angular-spatial dependencies of light field data.

*Table 11.* **Robustness analysis against sensor failures.** We simulate sensor malfunctions by randomly dropping the auxiliary modality views with varying probabilities from 0% to 90%. The SOTA methods experience severe performance degradation as the missing rate increases. Conversely, our method demonstrates considerable stability and graceful degradation, validating that the LF-ECMA module effectively preserve geometric integrity even under extreme data sparsity.

| Method | Modality Missing Rate | | | | | |
|---|---|---|---|---|---|---|
| | 0% | 10% | 30% | 50% | 70% | 90% |
| CMX (Zhang et al., 2023a) | 77.57 | 75.84 (-1.73) | 72.86 (-4.71) | 69.51 (-8.06) | 67.45 (-10.12) | 63.98 (-13.59) |
| CMNeXt (Zhang et al., 2023b) | 80.42 | 79.65 (-0.77) | 77.47 (-2.95) | 73.41 (-7.01) | 70.38 (-10.04) | 66.54 (-13.88) |
| MMSFormer (Reza et al., 2024) | 81.19 | 78.46 (-2.73) | 77.98 (-3.21) | 75.48 (-5.71) | 71.54 (-9.65) | 68.38 (-12.81) |
| without VRWKV & EPI Bias | 80.20 | 79.61 (-0.59) | 77.57 (-2.63) | 76.45 (-3.75) | 73.64 (-6.56) | 71.52 (-8.68) |
| **Ours** | **86.46** | **85.92 (-0.54)** | **84.85 (-1.61)** | **83.02 (-3.44)** | **81.45 (-5.01)** | **80.77 (-5.69)** |

learned from RGB images. Consequently, the Baseline tends to perform a superficial adaptation, relying primarily on texture cues rather than learning the complex angular-spatial dependencies intrinsic to light field data.

In contrast, our $E^2I$-VRWKV demonstrates a substantial Feature Reorganization, characterized by a significantly flattened distribution with a wider dynamic range. This transformation is directly attributed to the EPI Geometric Prior Generator, which forces the network to reconstruct its latent space to accommodate high-dimensional geometric information. By explicitly modeling the EPI structure, our method effectively drives the backbone to internalize the physical laws of the light field, thereby enhancing its discriminative power for depth discontinuities and occlusion boundaries compared to the texture-dominated Baseline.

## D. Robustness Analysis against Sensor Failures.

Real-world light field acquisition systems are highly susceptible to sensor failures, occlusions, or transmission bandwidth limits. To verify the reliability of our model under such non-ideal conditions, we simulate sensor malfunction scenarios by randomly dropping auxiliary angular views with probabilities ranging from 0% to 90%. Specifically, we implement this simulation by replacing the pixel values of the selected dropped views with all-zero tensors. This operation effectively renders the inputs as fully black images, thereby completely removing any visual information from those specific angles while maintaining the input tensor shape. Table 11 reports the quantitative results, from which we draw two key observations regarding the robustness of the model and the effectiveness of our proposed modules.

**Competitiveness over SOTA Methods.** As the modality missing rate increases, existing SOTA methods such as

CMX (Zhang et al., 2023a), CMNeXt (Zhang et al., 2023b), and MMSFormer (Reza et al., 2024) exhibit a precipitous decline in performance. Specifically, CMNeXt suffers a sharp drop of over 13% in mIoU at a 90% missing rate. This vulnerability stems from their fusion strategies, which typically treat angular views as supplementary feature channels; once these channels are blocked, the information flow is severed. In contrast, $E^2I$-VRWKV demonstrates considerable graceful degradation. Even with 90% of the auxiliary views missing, our method maintains an mIoU of 80.77%, which surpasses the full-modality performance of 77.57% achieved by CMX and is comparable to the 80.42% performance of CMNeXt. This indicates that our model does not merely read the inputs but internalizes the underlying light field physics, allowing it to reconstruct and recover the geometric structure even from extremely sparse signals.

**Effectiveness of Proposed Modules.** Comparing our full model with the Baseline, which excludes the EPI Geometric Prior and Interaction-Aware VRWKV modules, further validates the contribution of our components to system robustness. While the Baseline achieves a decent mIoU of 80.20% at 0% missing, its performance drops significantly to 71.52% under the 90% missing scenario, representing a degradation gap of 8.68%. However, with the integration of our proposed modules, this degradation gap is narrowed to just 5.69%, where the performance decreases from 86.46% to 80.77%. The widening performance gap between Ours and Baseline as the missing rate increases, from 6.26% at 0% missing to 9.25% at 90% missing, provides strong empirical evidence that the Explicit EPI Prior and VRWKV modules act as a geometric stabilizer. They effectively compensate for missing data by inferring global angular dependencies, thereby ensuring consistent segmentation accuracy in the face of sensor failures.

