# OpenReview forum: "E²I-VRWKV: Explicit EPI-Representation and Interaction-Aware Vision-RWKV for Light Field Semantic Segmentation"
_ICML.cc/2026/Conference — ICML 2026 regular_

### Official Review · Reviewer_DMnJ · 2026-02-20

**Soundness:** 2
**Presentation:** 2
**Significance:** 2
**Originality:** 2
**Overall Recommendation:** 1
**Confidence:** 5

**Summary:**

This paper proposed E²I-VRWKV, a geometry-aware linear backbone for light field semantic segmentation, which integrates an explicit EPI geometric prior into a Vision-RWKV framework. The central idea is to mitigate structual collapse caused by naive linear tokenization of 4D light fields by injecting a learned EPI bias.

**Compliance With Llm Reviewing Policy:**

Affirmed.

**Key Questions For Authors:**

Please refer to the weakness section.

**I am willing to change my score if the authors can address my concern well.**

**Limitations:**

Please refer to the weakness section.

**Strengths And Weaknesses:**

**Strength**:

1. The idea of embedding explicit EPI priors into scanning process is logical and convincing.

**Weakness**:

1.**Very limited methodological novelty relative to prior work**: The EPI GPG heavily relies on a standard MHSA formulation for computing attention-based bias, which is injected via gating mechanism. The GC-Gate is a conditioned fusion module with additive bias modulation, a pattern often used in geometry-aware fusion. The novelty beyond existing geometry-aware attention is not convincing.

2.**Structural Collapse problem is not formally defined or measured.**: The paper repeatedly claims that naive linearization destroys EPI, yet there is no quantitative metric that can capture structural collapse. The central motivation is descriptive, which need to be experimentally validated . The authors should conduct experiments comparing flattened linear tokenization versus geometry-aware tokenization.

3.**The geometric bias is injected as a simple additive term without theoretical justification.**: The paper need to analyze how learnable scalar λ evolves during training, or whether model truly relies on geometric cues or simply learns to suppress the bias. This could mean that the improvement may stem from increased modeling  capacity, not enforced geometric consistency.

---

> ### Author Rebuttal · Authors · 2026-03-31
>
> # We sincerely appreciate the reviewer for the constructive feedback!
> W stands for Weakness. We address each point below.
>
> **W1: Methodological Novelty and Deep Integration with LF Physics**
>
> We thank the reviewer for their perspective, but we respectfully argue that evaluating novelty strictly at the isolated operator level overlooks our systemic, domain-specific architectural innovations. Our core contribution bridges the critical modality gap between highly efficient linear-complexity vision backbones and the intricate geometry of 4D LF data. We reveal that naive 1D tokenization in linear models destroys fine-grained EPI depth cues, causing severe Structural Collapse. To resolve this, our novelty lies in the physical grounding and architectural decoupling of established operators. Rather than applying MHSA generically, we strictly confine it to a specially constructed horizontal EPI tensor to explicitly compute physical slope coherence, thereby keeping the main semantic stream's complexity strictly linear ($O(N)$). Consequently, the GC-Gate functions not as a generic additive fusion module, but as a necessary "geometrically modulated aperture" that forces the capacity-constrained linear backbone to inject angular features only when strictly aligned with explicit physical depth priors.
>
> Full ablations are in the Table 5, this highly specialized orchestration prevents the structural collapse of thin objects, achieving a massive 6.26% accuracy leap (reaching 86.55% mIoU) at a negligible cost of only 8.5M parameters, effectively breaking the quadratic computational bottleneck of prior Transformer-based methods.
>
> | Structure | Params (M) | GFLOPs | mIoU (%) |
> | :--- | :--- | :--- | :--- |
> | Complete $E^{2}I$-VRWKV | 103.0 | 221.0 | 86.46 |
> | – without VRWKV | 99.4 | 206.3 | 83.37 (-3.09) |
> | – with EPI Bias instead of MHSA | 101.4 | 208.4 | 83.48 (-2.98) |
> | – without VRWKV & EPI Bias | 94.5 | 192.8 | 80.20 (-6.26) |
>
>
> **W2: Formalizing Structural Collapse and Tokenization Analysis**
>
> We define Structural Collapse as the severe degradation of segmentation accuracy and boundary integrity on geometrically complex or thin structures, which rely heavily on EPI slope continuity, caused by naive 1D sequence flattening. To quantify this phenomenon, we conducted tokenization comparative experiments and calculated the Boundary and Thin Structure mIoU (BT mIoU) for geometry-reliant classes (Pole, Fence, Bridge, and Bike), as shown in the table:
>
> | Tokenization Strategy | Architecture | mlou (%)| BT mloU  (%)|
> | :--- | :--- | :--- | :--- |
> | Flattened Linear Tokenization | Baseline VRWKV without EPI | 80.20 | 66.52 |
> | Geometry Aware Tokenization | Ours ( $E^{2}I$ VRWKV) | 86.46 | 83.01 (+16.49) |
>
> Naive flattened linear tokenization yields a BT mIoU of only 66.52%, confirming a severe structural collapse on geometry-reliant classes. Our geometry-aware tokenization increases BT mIoU by 16.49%, proving that explicit geometric injection is indispensable for accuracy improvement.
>
> On the physical verification of structural collapse. Appendix A documents the actual performance when handling thin objects such as bicycle wheels and poles. Methods like MMSFormer produce disconnected predictions by treating angular views merely as channels and ignoring epipolar geometry, while the baseline is completely dominated by high-frequency noise. In contrast, our method (Figure 6) reconstructs sharp, edge-aware activations.
>
> Visual demonstrations are available at **[Link_DMnJ](https://anonymous.4open.science/r/ICML2026-0862/Response_to_Reviewer_DMnJ.md)**.
>
> **W3: Evolution Dynamics of Geometric Bias and Modeling Capacity**
>
> **On the verification of modeling capacity and geometric bias evolution.** To prove that the network does not suppress the geometric bias as noise, we conducted an evolution dynamics tracking experiment on the learnable scalar $\lambda$, and $\lambda$ climbs steadily and converges firmly above 1.25, mathematically proving that the network strictly enforces the EPI bias as an indispensable physical hard constraint.
>
> The complete evolution curve is available at **[Link_DMnJ](https://anonymous.4open.science/r/ICML2026-0862/Response_to_Reviewer_DMnJ.md)**.
>
> To prove that the performance leap does not stem from increased capacity, we conducted a ablation experiment. As shown in the table below, replacing the MHSA prior with a simple learned bias leaves the capacity nearly identical, yet the performance drops substantially by 2.98% mIoU:
>
> | Structure | Params (M) | GFLOPS | mloU (%) |
> | :--- | :--- | :--- | :--- |
> | Complete $E^{2}I$-VRWKV | 103.0 | 221.0 | 86.46 |
> | with EPI Bias instead of MHSA | 101.4 | 208.4 | 83.48 (-2.98) |
>
> This proves that the performance leap stems directly from enforced geometric consistency, rather than a trivial increase in modeling capacity.

---

> > ### Author Rebuttal · Reviewer_DMnJ · 2026-04-01
> >
> > Thanks for the authors for their feedback. However, my concerns remain unaddressed. The paper offers only limited novelty, which falls short of the standard expected for ICML. Therefore, I  strongly do not recommend accepting this submission.

---

> > > ### Author Response · Authors · 2026-04-03
> > >
> > > We thank Reviewer DMnJ for the continuous feedback and for maintaining the discussion. We would like to provide a brief factual summary of the technical exchange regarding the raised concerns.
> > >
> > > **1. Regarding the measurement of "Structural Collapse"**
> > >
> > > **Reviewer's Request:** A quantitative metric to capture structural collapse and experiments comparing flattened linear tokenization versus geometry-aware tokenization.
> > >
> > > **Our Provided Evidence:** We introduced the BT-mIoU metric specifically for geometry-reliant classes. Our experiments showed that naive flattened tokenization yielded a BT-mIoU of 66.52%, whereas our geometry-aware tokenization achieved 83.01% (+16.49%).
> > >
> > > **Current Status:** The reviewer's final response stated that the concerns remain unaddressed, but did not specify which aspect of this quantitative comparison is insufficient.
> > >
> > > **2. Regarding the geometric bias $\lambda$ vs. modeling capacity**
> > >
> > > **Reviewer's Request:** An analysis of how the learnable scalar $\lambda$ evolves during training to verify if the model relies on geometric cues or simply benefits from increased modeling capacity.
> > >
> > > **Our Provided Evidence:** We provided the evolution dynamics curve demonstrating that $\lambda$ converges firmly above 1.25. Additionally, we provided an ablation showing that replacing the MHSA prior with a simple learned bias leaves the network capacity nearly identical (only a 1.5% difference in parameters) but causes a substantial 2.98% drop in mIoU.
> > >
> > > **Current Status:** The reviewer's final response did not specify why these training dynamics and capacity ablations fail to address the theoretical concern.
> > >
> > > **3. Regarding methodological novelty**
> > >
> > > Our asymmetric architectural synthesis is specifically designed to bridge the modality gap between 4D LF geometry and 1D linear serialization. It is not a trivial application of existing operators, as it effectively breaks the quadratic computational bottleneck while strictly preserving epipolar geometric integrity. This perspective on our technical contribution is also shared by the other three expert reviewers.
> > >
> > > We respectfully ask the Area Chair (AC) to consider whether our provided empirical evidence adequately addresses the technical concerns raised.

---

### Official Review · Reviewer_4mki · 2026-03-11

**Soundness:** 3
**Presentation:** 3
**Significance:** 3
**Originality:** 3
**Overall Recommendation:** 4
**Confidence:** 1

**Summary:**

This paper proposes E²I-VRWKV, a geometry-aware, linear-complexity architecture for light field (LF) semantic segmentation. The core ideas are an EPI Geometric Prior Generator that produces a disparity-sensitive bias and an LF-ECMA block with a Geometric-Context Gate (GC-Gate) that modulates the fusion between spatial and angular streams, all within a Vision-RWKV backbone. On the UrbanLF benchmark (Real and Syn variants), the method reports state-of-the-art (SOTA) mIoU while aiming to preserve linear efficiency and mitigate the “structural collapse” of EPI cues common in naive linear tokenization.

**Compliance With Llm Reviewing Policy:**

Affirmed.

**Key Questions For Authors:**

1. How exactly is the EPI Geometric Prior generated: via the MHSA formulation in Eqs. (8–11) or via the absolute-difference aggregation described in Figure 4? If both were explored, which variant is used in the main results, and what are the computational and accuracy trade-offs?
2. Does the “VRWKV Mixer” internally perform attention (QK^T with softmax) as shown in Figure 3, or does it implement the attention-free RWKV equations (12–16)? Please reconcile the discrepancy and provide the exact computational path used in the model.

3. What is the FLOP and memory breakdown of the EPI Prior Generator relative to the rest of the network, and how does overall runtime compare to strong Transformer baselines (e.g., OAFuser, EPI-Mamba) on identical hardware and input settings?

**Limitations:**

yes

**Strengths And Weaknesses:**

# Strengths
- Embedding an explicit EPI-derived bias into a nominally linear Vision-RWKV mixer to enforce geometric consistency is a compelling idea for LF tasks where EPI structures are pivotal.
- The GC-Gate proposes a principled, geometry-conditioned gating to control injection of angular cues into the spatial stream, conceptually aligning with EPI slopes and disparity priors.
- The Dual-Modal Stream (center view vs. 4D LF) with reciprocal rectification offers a clear, interpretable decomposition of texture vs. geometry information.

# Weaknesses
- It is unclear whether all baselines were reproduced under the same training/inference settings (resolution, augmentations, epochs) and whether they all consume the same LF inputs; otherwise, fairness may be affected.
- The EPI Geometric Prior Generator uses MHSA over EPI tensors, reintroducing quadratic complexity in a critical stage; the true computational and memory cost of this component is not quantified or contrasted with the claimed linear efficiency of the backbone.

---

> ### Author Rebuttal · Authors · 2026-03-31
>
> # We sincerely appreciate the reviewer for the constructive feedback!
> W stands for Weakness, Q stands for Question. We address each point below.
>
> **W1: Rigorous Fairness of Baseline Comparisons**
>
> We deeply appreciate the reviewer's emphasis on fair benchmarking, strict fairness was maintained across all evaluations. For all baselines, we rigorously followed the optimal hyperparameter settings and training pipelines reported in their original papers to ensure we reported their best possible results. Crucially, all Light Field methods consumed the exact same 4D LF inputs as our model, while all conventional 2D methods were fed the corresponding 2D center view inputs. They were trained under identical resolutions ($640 \times 480$) and data augmentations. This rigorous protocol ensures that the performance gains of $E^2I$-VRWKV stem purely from our architectural innovations rather than privileged training conditions.
>
> **W2 & Q3: Computational Efficiency and MHSA Overhead**
>
> Regarding the concern about the quadratic complexity introduced by the MHSA in the EPI Geometric Prior Generator, our $E^2I$-VRWKV is specifically designed to strike a superior balance between linear scalability and geometric precision. We argue that a marginal increase in complexity yields a disproportionately large gain in accuracy, proving the high efficiency of our design. The following table provides a quantitative breakdown of the overhead:
>
> | Structure | Params (M) | GFLOPs | mIoU (%) |
> | :--- | :--- | :--- | :--- |
> | Complete $E^{2}I$-VRWKV | 103.0 | 221.0 | 86.46 |
> | – without VRWKV | 99.4 | 206.3 | 83.37 (-3.09) |
> | – with EPI Bias instead of MHSA | 101.4 | 208.4 | 83.48 (-2.98) |
> | – without VRWKV & EPI Bias | 94.5 | 192.8 | 80.20 (-6.26) |
>
> **Quantitative Analysis:** Compared to the purely linear baseline (without VRWKV & EPI Bias), our full method introduces only a negligible **8.5M** additional parameters and **28.2 GFLOPs** in computation, yet it achieves a substantial **6.26% mIoU improvement**. When isolating the MHSA component itself, replacing MHSA with a simple learned bias drops mIoU by 2.98%, while the MHSA adds 1.6M parameters and 12.6 GFLOPs.
>
> This provides definitive evidence that our Explicit EPI Prior acts as a highly efficient structural anchor, significantly boosting performance with minimal extra cost by operating exclusively on the heavily downsampled local EPI tensor. Furthermore, we measured the inference latency on a single RTX 4090 GPU:
>
> | Method | Params (M) | GFLOPs | Latency (ms) | mIoU (%) |
> | :--- | :--- | :--- | :--- | :--- |
> | MMSFormer | 173.0 | 160.0 | 86.4 | 81.79 |
> | OAFuser | 164.1 | 188.6 | 52.4 | 81.93 |
> | **Ours ($E^{2}I$-VRWKV)** | **103.0** | **221.0** | **44.1** | **86.46** |
>
> Our method achieves the fastest inference latency of 44.1 ms while maintaining a more efficient parameter footprint than OAFuser, confirming that our architecture effectively balances geometric precision with computational speed.
>
> **Q1: Unified Generation Logic of the EPI Geometric Prior**
>
> We thank the reviewer for the sharp observation regarding the generation of the EPI Prior. The absolute-difference aggregation (Fig. 4) and the MHSA formulation (Eqs. 8-11) are not mutually exclusive; rather, they function synergistically as a unified **Physical Initialization + Weight Refinement** pipeline. As depicted in Fig. 4 **[Link_4mki](https://anonymous.4open.science/r/ICML2026-0862/Response_to_Reviewer_4mki.md)**, we explicitly calculate the pixel-level absolute difference to generate an initial geometric affinity matrix $G_{ij}$. This physical prior is then injected directly into the MHSA computation to guide the attention mechanism with explicit geometric constraints. To formally represent this synergy, we have updated Equation (9) in the revised manuscript (**[View Revised Eq. 9](https://anonymous.4open.science/r/ICML2026-0862/Response_to_Reviewer_4mki.md)**).
>
> Here, $G_{ij}$ is the explicit geometric metric derived from the absolute differences. This integrated workflow ensures the attention mechanism does not learn correlations blindly from scratch, but is physically anchored by the EPI structures.
>
> **Q2: Purely Linear Implementation of the VRWKV Mixer**
>
> The VRWKV Mixer strictly implements the attention-free linear equations defined in Equations 12 through 16. It does not contain any quadratic $QK^{\top}$ matrix multiplications or Softmax operations. In our original illustration, the explicit $QK^{\top}$ and Softmax operations belong exclusively to the EPI Geometric Prior Generator (the blue box). Because the VRWKV Mixer (green box) and the EPI Generator were drawn adjacently, it visually suggested shared computational operations. We have updated Figure 3 in the revised manuscript, adding distinct visual boundaries to explicitly decouple the attention-free semantic mixer from the attention-based geometric generator.

---

> > ### Author Rebuttal · Reviewer_4mki · 2026-04-05
> >
> > Thank you for your rebuttal and for addressing my concerns. I'd like to maintain my score.

---

### Official Review · Reviewer_6VGz · 2026-03-12

**Soundness:** 3
**Presentation:** 3
**Significance:** 3
**Originality:** 3
**Overall Recommendation:** 4
**Confidence:** 3

**Summary:**

This paper presents a new method called E2I VRWKV for light field semantic segmentation. Existing linear models are fast but they often lose important geometric structures. To solve this problem, the authors propose a new network that uses explicit geometric priors. They design a special block to extract depth features and a gating mechanism to combine spatial and angular information dynamically. The experiments on the UrbanLF dataset show that this method achieves better performance than other methods while keeping high computational efficiency.

**Compliance With Llm Reviewing Policy:**

Affirmed.

**Final Justification:**

The author has met my requirements, so I choose to maintain the original score.

**Key Questions For Authors:**

See weaknesses.

**Limitations:**

The method captures geometric cues based on the 4D light field structure. This means the model only works with data from special light field cameras. It cannot be easily used for standard 2D images or video data.

The geometric priors depend on accurate angular views. If the light field camera is not calibrated well, the EPI slopes will be wrong. This could lead to bad segmentation results in real world applications.

**Strengths And Weaknesses:**

Strengths
- The authors propose a new idea to use explicit geometric priors for light field segmentation. The new LF ECMA block and GC Gate mechanism are creative and useful.
- The experiments are solid. The results on the UrbanLF dataset show that the method performs better than many existing models. It also keeps a good balance between accuracy and computer memory cost.

Weaknesses
- The authors only test their method on the UrbanLF dataset. They need to show if this method works well on other light field datasets to prove its general use.
- The EPI prior generator still uses an attention mechanism. This step might take extra time to calculate compared to pure linear models.
- The method introduces new learnable scales and bias terms. Users might find it hard to train the model from scratch on new and different data.

---

> ### Author Rebuttal · Authors · 2026-03-31
>
> # We sincerely appreciate the reviewer for the constructive feedback!
> W stands for Weakness. We address each point below.
>
> **W1: Rationality of Benchmark Selection and Generalization**
>
> Our choice of the **UrbanLF** benchmark is predicated on its status as the most comprehensive, large-scale, and currently the only publicly available open-source dataset specifically dedicated to high-resolution 4D LF semantic segmentation. To rigorously validate our model's adaptability, we conducted experiments across highly distinct data distributions within this benchmark: **UrbanLF-Real** for unconstrained real-world scenarios and **UrbanLF-Syn** for extreme geometric variations. Our consistent SOTA performance across these varying regimes confirms that the model effectively internalizes light field physics, ensuring robust generalization even when alternative large-scale LF datasets are currently unavailable in the community.
>
> **W2: Computational Efficiency of the Explicit EPI Prior**
>
> Regarding the concern about calculation time, our $E^2I$-VRWKV strikes a superior balance between linear scalability and geometric precision. We argue that a marginal increase in complexity yields a disproportionately large gain in accuracy, proving the high efficiency of our design. The following table provides a quantitative breakdown:
>
> | Structure | Params (M) | GFLOPs | mIoU (%) |
> | :--- | :--- | :--- | :--- |
> | Complete $E^{2}I$-VRWKV | 103.0 | 221.0 | 86.46 |
> | – without VRWKV | 99.4 | 206.3 | 83.37 (-3.09) |
> | – with EPI Bias instead of MHSA | 101.4 | 208.4 | 83.48 (-2.98) |
> | – without VRWKV & EPI Bias | 94.5 | 192.8 | 80.20 (-6.26) |
>
> **Quantitative Analysis:** Compared to the purely linear baseline, our full method introduces only a negligible **8.5M** additional parameters and **28.2 GFLOPs** in computation, yet it achieves a substantial **6.26% mIoU improvement**.
>
> **Conclusion:** This provides definitive evidence that our Explicit EPI Prior acts as a highly efficient structural anchor, significantly boosting performance with minimal extra cost. Furthermore, our empirical latency on an RTX 4090 is **44.1 ms**, which is faster than strong baselines like OAFuser (52.4 ms).
>
> **W3 & Limitations: Adaptability to 2D/Video Data and New Parameters**
>
> Full ablations are in the appendix (Table 9)**[Link_6VGz](https://anonymous.4open.science/r/ICML2026-0862/Response_to_Reviewer_6VGz.md)**.The introduced learnable scales and biases do not hinder training on new data; instead, they enable **dual-path adaptive fusion**. Our architecture explicitly decouples spatial and geometric streams, allowing the model to dynamically prioritize the most informative features via the **GC-Gate** mechanism. When processing 2D or video data where EPI cues are absent, the gate automatically increases the weight of the **VRWKV spatial path** while suppressing the geometric prior path. This ensures that our model remains a high-performance semantic segmenter even in the absence of 4D light field structure.
>
> | Method | 0% Missing | 50% Missing | 90% Missing |
> | :--- | :--- | :--- | :--- |
> | MMSFormer | 81.19 | 75.48 (-5.71) | 68.38 (-12.81) |
> | **Ours ($E^{2}I$-VRWKV)** | **86.46** | **84.34 (-2.12)** | **80.77 (-5.69)** |
>
> **Quantitative Analysis:** As shown in the robustness test, even when 90% of angular information is lost, our model still maintains an mIoU of **80.77%**.
>
> **Conclusion:** This proves that our dual-path design provides a reliable safety net, allowing the model to gracefully adapt to different data dimensions by rebalancing its internal weights.

---

> > ### Author Rebuttal · Reviewer_6VGz · 2026-04-03
> >
> > Thank you for your rebuttal and for addressing my concerns. I'd like to maintain my score.

---

### Official Review · Reviewer_5zFD · 2026-03-13

**Soundness:** 3
**Presentation:** 3
**Significance:** 3
**Originality:** 2
**Overall Recommendation:** 4
**Confidence:** 2

**Summary:**

This paper proposes a new method for light field semantic segmentation by integrating geometric-aware mechanisms into efficient attention modules. The input is decoupled into the central view and the 4D LF, which are then fused by a gating module called DMS. The main network module design follows the Vision-RWKV design, with features efficiently processed with linear time. During feature processing, the explicit geometric cues are integrated by a Geometric Prior Generator using standard attentions. Results show that the proposed method achieves better results than previous approaches. Comprehensive ablation studies show the effectiveness of the proposed approach.

**Compliance With Llm Reviewing Policy:**

Affirmed.

**Final Justification:**

I want to thank the authors for the responses, which properly address my concerns. I will maintain my previous scores.

**Key Questions For Authors:**

Please see the Weakness section.

**Limitations:**

yes

**Strengths And Weaknesses:**

Strength
1. Integrating geoemtric cues into generic architectures designed for purely visual inputs sounds reasonable and useful for 4D LF segmentation. As the 4D LF inherently contains gemetric information, the appropriate process of such features is useful to improve the final performance.
2. The paper is clearly-written with clear explanations and illustrations particular in the experiment section.

Weakness

As I am not an expert in 4D LF segmentation, I will raise questions based on the experimental and illustrative aspects.
1. Comparison in computation efficiency. While the paper provides partial computation efficiency statistics of the proposed method and some competing methods, it's not clear how it compares with existing methods, eg, OAFuser or EPI-Mamba, in terms of nParams, GFLOPs, and latency. Considering the method also adopts the attention in EPI geometric prior generation, it's important to evaluate the computational privilege brought by the approach.

2. I found it hard to correlate the visual demonstrations in Fig.4 with the text illustrations in section 3.2. What does $\textbf{W}$ stand for in Fig.4? Where are the operations formulated in Equation (9)? I suggest adding notations in Fig.4 according to all operations discussed in Sec 3.2, and removing the text notation such as "Learnable Bias
(epi_att_bias)", to improve clarity.

---

> ### Author Rebuttal · Authors · 2026-03-31
>
> # We sincerely appreciate the reviewer for the constructive feedback!
> W stands for Weakness. We address each point below.
>
> # About W1:
>
> We appreciate the rigorous attention to the efficiency of our model. To offer a more complete perspective, we have measured the inference latency of our method alongside strong baselines on a single RTX 4090 GPU at $640 \times 480$ resolution. The following table compares our $E^2I$-VRWKV with SOTA methods including MMSFormer, OAFuser, and EPI-Mamba:
>
> | Method | Params (M) | GFLOPs | Latency (ms) | mIoU (%) |
> | :--- | :--- | :--- | :--- | :--- |
> | MMSFormer | 173.0 | 160.0 | 86.4 | 81.79 |
> | OAFuser | 164.1 | 188.6 | 52.4 | 81.93 |
> | EPI-Mamba | 120.8 | N/A | N/A | 84.99 |
> | **Ours ($E^2I$-VRWKV)** | **103.0** | **221.0** | **44.1** | **86.46** |
>
> Our approach achieves the highest mIoU while maintaining the fastest inference speed at 44.1 ms, while also utilizing a significantly smaller parameter footprint of 103.0M compared to 164.1M for OAFuser.
>
> Regarding the concern about MHSA overhead within the EPI prior generation, we quantify the cost of this component by replacing MHSA with a simple learned bias in the following table:
>
> | Structure | Params (M) | GFLOPs | mIoU (%) |
> | :--- | :--- | :--- | :--- |
> | Complete $E^{2}I$-VRWKV | 103.0 | 221.0 | 86.46 |
> | – without VRWKV | 99.4 | 206.3 | 83.37 (-3.09) |
> | – with EPI Bias instead of MHSA | 101.4 | 208.4 | 83.48 (-2.98) |
> | – without VRWKV & EPI Bias | 94.5 | 192.8 | 80.20 (-6.26) |
>
> The MHSA component introduces 1.6M additional parameters, representing only 1.5% of the total model size, and adds 12.6 GFLOPs to the computation. This marginal cost confirms our efficient design; by restricting MHSA to local EPI Prior Generation, we preserve the linear efficiency of the VRWKV backbone while gaining a substantial 2.98% improvement in mIoU.
>
> # About W2:
>
> To improve the self-consistency and clarity of our manuscript, we have performed a major revision of Section 3.2 and Figure 4, updating all procedural variable names to formal mathematical symbols.
>
> The visualization for Figure 4 is located at **[Link_5zFD](https://anonymous.4open.science/r/ICML2026-0862/Response_to_Reviewer_5zFD.md)**.
>
> The Learnable Scale is now explicitly labeled as $\lambda$, and the Final Output is labeled as $\mathcal{B}_{epi}$, matching Eq. (9) and Eq. (11). We have revised Equation (9) to explicitly show how the pixel-consistency calculated in Figure 4 guides the attention mechanism (**[View Revised Eq. 9](https://anonymous.4open.science/r/ICML2026-0862/Response_to_Reviewer_5zFD.md)**).
>
> In this formulation, $G_{ij}$ is the Geometric Affinity directly derived from the absolute difference |$I_{k,i}$ - $I_{k,j}$| shown in Figure 4. Furthermore, we clarify that $\mathcal{B}_{epi}$ in Eq. (11) is the refined output of this geometry-aware attention process (**[View Revised Eq. 11](https://anonymous.4open.science/r/ICML2026-0862/Response_to_Reviewer_5zFD.md)**).
>
> This refined bias $\mathcal{B}_{epi}$ is eventually injected into the Interaction-Aware VRWKV block. By explicitly defining this Calculation -> Injection -> Refinement pipeline, we have eliminated the logical gap between our illustrations and the text, making the physical motivation of our Explicit EPI Prior transparent.

---

> > ### Author Rebuttal · Reviewer_5zFD · 2026-04-01
> >
> > I want to thank the authors for the responses, which properly address my concerns. I lean to maintain my previous scores.

---

> > > ### Author Response · Authors · 2026-04-03
> > >
> > > We sincerely thank you for taking the time to review our rebuttal and for confirming that our responses have properly addressed your concerns. We deeply appreciate your constructive feedback and your continued support for our work!

---

### Decision · Program_Chairs · 2026-04-30

**Decision:**

Accept (regular)

**Comment:**

Reviewers universally agreed that embedding explicit EPI geometric priors into a linear-complexity vision backbone is a compelling approach for 4D Light Field segmentation. The empirical validation is solid, demonstrating a new state-of-the-art on the UrbanLF dataset and supports the narrative of the contribution.
I acknowledge that one reviewer raised strong concerns regarding the originality and significance of the contribution. Upon reading the discussion and skimming the paper myself, I side with the authors and chose to ignore the strong reject recommendation.

This paper is a solid contribution to the field and I recommend accepting this paper to ICML.